# Digital and online symptom checkers and health assessment/triage services for urgent health problems: systematic review

Duncan Chambers,  Anna J Cantrell, Maxine Johnson, Louise Preston, Susan K Baxter, Andrew Booth, Janette Turner

School of Health and Related Research, The University of Sheffield, Sheffield, UK

**Correspondence to**
Mr Duncan Chambers;
d.chambers@sheffield.ac.uk

## ABSTRACT

**Objectives** In England, the NHS111 service provides assessment and triage by telephone for urgent health problems. A digital version of this service has recently been introduced. We aimed to systematically review the evidence on digital and online symptom checkers and similar services.

**Design** Systematic review.

**Data sources** We searched Medline, Embase, the Cochrane Library, Cumulative Index to Nursing and Allied Health Literature (CINAHL), Health Management Information Consortium, Web of Science and ACM Digital Library up to April 2018, supplemented by phrase searches for known symptom checkers and citation searching of key studies.

**Eligibility criteria** Studies of any design that evaluated a digital or online symptom checker or health assessment service for people seeking advice about an urgent health problem.

**Data extraction and synthesis** Data extraction and quality assessment (using the Cochrane Collaboration version of QUADAS for diagnostic accuracy studies and the National Heart, Lung and Blood Institute tool for observational studies) were done by one reviewer with a sample checked for accuracy and consistency. We performed a narrative synthesis of the included studies structured around pre-defined research questions and key outcomes.

**Results** We included 29 publications (27 studies). Evidence on patient safety was weak. Diagnostic accuracy varied between different systems but was generally low. Algorithm-based triage tended to be more risk averse than that of health professionals. There was very limited evidence on patients' compliance with online triage advice. Study participants generally expressed high levels of satisfaction, although in mainly uncontrolled studies. Younger and more highly educated people were more likely to use these services.

**Conclusions** The English 'digital 111' service has been implemented against a background of uncertainty around the likely impact on important outcomes. The health system may need to respond to short-term changes and/or shifts in demand. The popularity of online and digital services with younger and more educated people has implications for health equity.

**PROSPERO registration number** CRD42018093564.

### Strengths and limitations of this study

► This systematic review was based on a rigorous search of the literature which maximised efficiency by combining an initial focused search with subsequent rounds of follow-up searching, including searches for named symptom checker systems.

► Our narrative synthesis approach used a mixture of description and tabulation to summarise the evidence, including the overall strength of the evidence base for each of the prespecified outcomes of interest.

► Given the decision to implement a national urgent care service based on digital symptom checkers in the National Health Service in England, our study highlights areas of uncertainty that will need to be resolved by research and data collection.

► The review inclusion criteria were relatively broad and findings from symptom checker systems for specific conditions may not be applicable to more general systems and vice versa.

► We have also included studies of symptom checkers as part of electronic consultation systems in general practice, which again represents a slightly different setting from a general 'digital 111' service, and this should be kept in mind when interpreting the results.

## INTRODUCTION

Digital and online symptom checkers and assessment services are used by patients seeking guidance about health problems, including some that may require urgent action. These services generally provide people with possible alternative diagnoses based on their reported symptoms and/or suggest a course of action (eg, self-care, make a general practitioner (GP) appointment or go to an emergency department (ED)).

In England, the NHS111 service provides assessment and triage by telephone for problems that are urgent but not classified as emergencies. The latest data from National Health

BMJ

Service (NHS) England[1] show that in September 2018 there were over 1.27 million calls to NHS111, an average of 42 400 per day. Outcomes of these calls were that 13.2% had ambulances despatched; 9.5% were recommended to attend an ED; 58.7% were recommended to attend primary care; 4.8% to attend another service and 13.8% were not recommended to attend another service (eg, their condition was considered suitable for self-care).

NHS England has recently introduced a digital platform to make NHS111 accessible via a website or smartphone app. A beta version of the service (referred to as 'NHS111 Online') is available at https://111.nhs.uk/ (accessed 1 April 2019). The 'digital 111' service is seen as key to reducing demand for the telephone 111 service, enabling resources to be redirected to supporting 'integrated urgent and emergency care systems' as outlined in the 'NHS 5 year Forward View' and its 2017 update 'Next Steps on the NHS 5 year Forward View'.[2 3]

There is an expectation that a digital 111 platform will help to manage demand and increase efficiency in the urgent and emergency care system, complementing the agenda of locally based sustainability and transformation partnerships which involve the health service and local government working together to integrate and coordinate care.[4] However, there is a risk of increasing demand, duplicating healthcare contacts (by increasing the number of potential access routes into the system) and providing advice that is not safe or clinically appropriate. For example, an evaluation of the NHS111 telephone service at four pilot sites and three control sites found that in its first year the service was not successful in reducing 999 emergency calls or in shifting patients from emergency to urgent care.[5] A recent study of 23 symptom checker algorithms providing diagnostic and triage advice that would form the basis of a digital 111 platform found deficiencies in both their diagnostic and triage capabilities (based on patient vignettes).[6]

In 2017, NHS England carried out pilot evaluations of different systems in four regions of England. The evaluations aimed to assess whether digital/online triage was acceptable to users and connected them to appropriate clinical care.[7] The full report of the evaluations was not yet published at the time of writing. The objective of this systematic review was to inform further development of the proposed digital platform by summarising and critiquing the previous research in this area, both from the UK and overseas. The overall research question was: for people seeking guidance about an urgent health problem, what is the effect of digital and online services designed to assess symptoms and signpost patients to appropriate services (compared with non-digital services or no comparator) on important clinical and health service outcomes? Outcomes include safety; clinical and cost effectiveness; diagnostic and triage accuracy; impact on service use; patient/carer satisfaction; compliance with advice received and outcomes related to equity and inclusion.

## METHODS

The review protocol is available from the project website (https://www.journalslibrary.nihr.ac.uk/programmes/hsdr/164717/).

### Literature search and screening

Initial scoping searches revealed that a highly sensitive search strategy, as typically conducted for systematic reviews, retrieved a disproportionately high number of references on GP decision-making and triage as demonstrated by the examination of sample search results (eg, first 100). We therefore devised a three-stage retrieval strategy as an acceptable alternative to comprehensive topic-based searching. This involved:

1. Targeted searches of precise high specificity terms in seven databases (Medline, Embase, the Cochrane Library, CINAHL, Health Management Information Consortium, Web of Science and ACM Digital Library). These searches were not restricted by language or date. The search strategies used for this part of the review are presented in online supplementary appendix 1.
2. Phrase searching for names of known symptom checkers using a list compiled from Semigran 2015 and other sources.
3. Citation searches and reference checking of key included studies and reviews, complemented by contact with service providers (directly and via websites).

The main literature search was completed in April 2018 and follow-up searches in May 2018. Search results were stored in a reference management system (EndNote) and imported into EPPI-Reviewer software for screening, data extraction and quality assessment. The search results were screened against the inclusion criteria by one reviewer, with a 10% random sample screened by a second reviewer. Uncertainties were resolved by discussion among the review team.

### Inclusion and exclusion criteria
#### Population

General population seeking information online or digitally to address an urgent health problem, including adults and children and issues arising from both acute and long-term chronic illness.

#### Intervention

Any online digital service designed to assess symptoms, provide health advice and direct patients to appropriate services. Services that only provide health advice were excluded, as were those that offer treatment, for example, online cognitive behavioural therapy (CBT) services.

#### Comparator

The 'gold standard' comparator is current practice of telephone assessment (eg, NHS111) or face-to-face assessment (eg, general practice, urgent care centre or ED). However, studies with other relevant comparators (eg, comparative performance in tests or simulations) or with no comparator were included if they addressed the research questions.

## Outcomes

The main outcomes of interest were safety (eg, any evidence of adverse events arising from following or ignoring advice from online/digital services); clinical effectiveness; costs/cost effectiveness; accuracy; impact on service use; compliance with advice received; patient/carer satisfaction and equity and inclusion. Accuracy covered (1) ability to provide a correct diagnosis and (2) ability to distinguish between high and low acuity/urgency problems (and hence direct patients to appropriate services).

## Study design

We did not restrict inclusion by study design (and included relevant audits or service evaluations in addition to formal research studies) but included studies had to evaluate (quantitatively or qualitatively) some aspect of an online/digital service.

## Other

Studies from any developed country healthcare system were eligible for inclusion.

## Excluded

Purely descriptive studies, conceptual papers, projections of possible future developments and studies conducted in low- or middle-income countries were excluded from the review.

## Data extraction and quality/strength of evidence assessment

We extracted and tabulated key data from the included studies, including study design, population/setting, results and key limitations. Data extraction was performed by one reviewer, with a 10% random sample checked for accuracy and consistency.

To characterise the included digital and online systems as interventions, we identified studies reporting on a particular system and extracted data from all relevant studies using a modification of the Template for Intervention Description and Replication checklist[8] which we designated Template for Intervention Description for Systems for Triage. Further details may be found in the full report.[9]

Quality (risk-of-bias) assessment was undertaken for peer-reviewed full publications only (ie, not grey literature publications (such as research reports, working papers or reports produced by government departments, academics, business and industry) or conference abstracts). Randomised controlled trials (RCTs) were assessed using the Cochrane Collaboration risk of bias tool. For diagnostic accuracy type studies, we used the Cochrane Collaboration version of QUADAS[10] and for other study designs we used the National Heart Lung and Blood Institute tool for observational cohort and cross-sectional studies (https://www.nhlbi.nih.gov/health-topics/study-quality-assessment-tools, accessed 25 March 2019). Quality assessment was performed by one reviewer, with a random 10% sample checked for accuracy and consistency.

Assessment of the overall strength (quality and relevance) of evidence for each research question is part of the narrative synthesis. Overall strength of the evidence base for key outcomes was assessed using an adaptation of the method described by Baxter et al.[11] This involves classifying evidence as 'stronger', 'weaker', 'conflicting' or 'insufficient' based on study numbers and design. Specifically, 'stronger evidence' represented generally consistent findings in multiple studies with a comparator group design or comparative diagnostic accuracy studies; 'weaker evidence' represented generally consistent findings in one study with a comparator group design and several non-comparator studies or multiple non-comparator studies; 'very limited evidence' represented an outcome reported by a single study; and finally, 'inconsistent evidence' represented an outcome where fewer than 75% of studies agreed on the direction of effect. All studies in the review, including those that did not meet criteria for risk-of-bias assessment, were included in the strength of evidence assessment.

## Evidence synthesis

We performed a narrative synthesis structured around the prespecified research questions and outcomes. We did not perform any meta-analyses because the included studies varied widely in terms of design, methodology and outcomes.

## Patient and public involvement (PPI)

The review was discussed at two meetings of a PPI group providing advice to the programme from which the review was commissioned (Sheffield HS&DR Evidence Synthesis Centre). At the meetings, there was discussion regarding the focus of the work, including a presentation on previous research on NHS111 telephone services to provide a context for understanding the current work. The meetings also included presentation and discussion of the findings of the review, in order to explore key messages for patients which could inform dissemination of the findings. Discussion during one meeting was structured using a strengths, weaknesses, opportunities and threats (SWOT) analysis approach, which revealed a number of potential concerns among patients (eg, reliability and consistency; high costs of programming and development; whether patients would follow advice given and threats to equity) as well as potential perceived benefits (eg, improved access to care at all hours; value to those who might feel embarrassed discussing their problem with a health professional). Involvement of the advisory group was beneficial in highlighting some issues that had also emerged from the systematic review, and enabled the reviewers to structure the review findings taking this into account. For example, the group's uncertainty about the likely impact of digital 111 was reflected in the review findings and recommendations for ongoing evaluation and further research. The review report also reflects the group's relatively cautious attitude (while recognising the need to update the way services are accessed) which

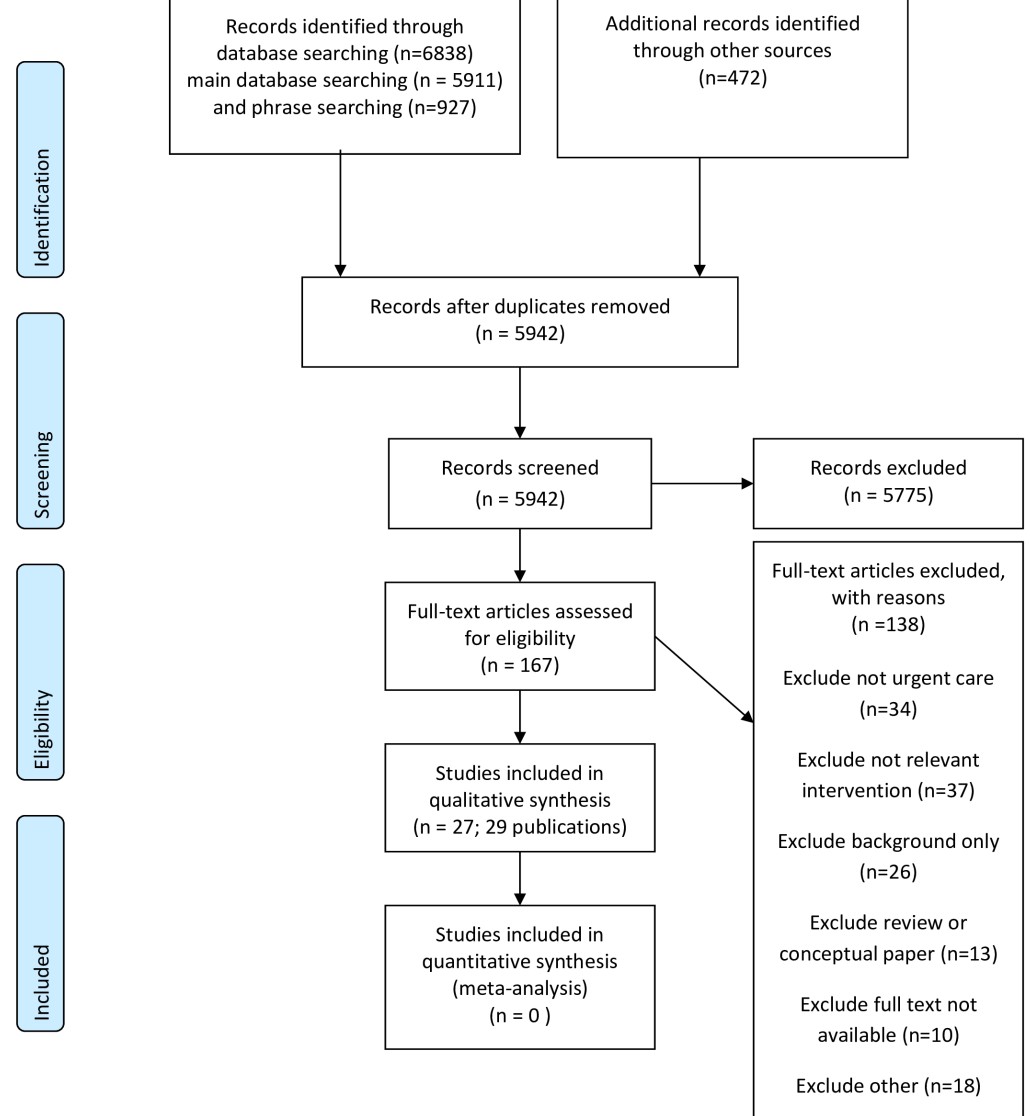

**Figure 1** Preferred Reporting Items for Systematic Reviews and Meta-Analyses flow diagram.

contrasts with the strong belief in some quarters that digital 111 will help to ensure that patients receive appropriate care more quickly while reducing 'inappropriate' visits to EDs and GP appointments.

## RESULTS
### Results of literature search
Twenty-seven studies (29 publications) were included in the review. Figure 1 presents the flow of studies through the selection process. Inter-rater agreement on initial study selection was moderate (Kappa=0.582). This reflects a degree of learning by the review team: our initial sift of the search results consciously favoured inclusivity and items found not to meet the inclusion criteria on detailed examination were subsequently discarded.

### Characteristics of included studies
Seventeen studies (table 1) evaluated symptom checkers as a self-contained intervention, of which eight covered a limited range of symptoms, for example, respiratory[12–14] or gastrointestinal[15 16] symptoms which we considered to be 'urgent'. The remaining studies in this group evaluated symptom checkers covering a wider range of common urgent care symptoms. Studies either evaluated a single system[17–20] or multiple systems.[6 21] We found only one study of a symptom checker specifically intended for the assessment of children's symptoms, a development of the Strategy for Off-Site Rapid Triage system for influenza-like illness.[22] Two reports with some overlap of content evaluated the 'babylon check' app.[17 23] Studies were conducted in the USA, UK or other European countries.

Five studies (four from the UK)[7 24–27] evaluated symptom checkers as part of a broader self-assessment and consultation system (often referred to as electronic

**Table 1** Studies of symptom checkers as a self-contained intervention

| Reference | Study design | System type | Comparator | Population/sample |
|---|---|---|---|---|
| Babylon Health[23] UK | • Uncontrolled observational No control group but some comparison with NHS111 telephone data | • Digital Smartphone app | • Health professional performance on real-world data • Other NHS111 data for 12 months from February 2017 | • General population Participants in the London pilot evaluation of 'digital 111' services |
| Berry et al[15] USA | • Simulation Evaluation of symptom checker performance on clinical vignettes | • Online 17 symptom checkers | • None | • Specific condition(s) Gastrointestinal symptoms |
| Berry et al[38] USA | • Controlled observational | • Online Three online symptom checkers (WebMD, iTriage and FreeMD) | • Health professional performance on real-world data | • Specific condition(s) Patients with a cough presenting to an internal medicine clinic |
| Berry et al[16] USA | • Controlled observational | • Online Three online symptom checkers (WebMD, iTriage, FreeMD) | • Health professional performance on real-world data | • Specific condition(s) Abdominal pain |
| Kellermann et al[12] USA | • Simulation The developed algorithm was tested against past patient records. | • Online Strategy for Off-Site Rapid Triage (SORT) was available on two interactive websites | • Health professional performance on real-world data The algorithm was tested against clinicians' decision on past patient records. | • Specific condition(s) Influenza symptoms |
| Little et al[13] UK | • Experimental Randomised controlled trial | • Online 'Internet Doctor' website | • Other Usual GP care without access to the Internet Doctor website | • Specific condition(s) Respiratory infections and associated symptoms |
| Luger et al[33] USA | • Simulation Described as 'human–computer interaction study' using think-aloud protocols. | • Online Google and WebMD | • Other Comparing two internet health tools. | • General population Older adults (50 years or older) |
| Marco-Ruiz et al[34] Norway | • Qualitative Qualitative element • Other 1. Online evaluation by users (problem detection) 2. Think aloud technique by smaller sample of participants (usability) | • Online Erdusyk | • None | • General population Internet tool users |
| Middleton et al[17] UK | • Simulation | • Digital 'babylon check' automatic triage system | • Health professional performance on test/ simulation Twelve ' clinicians' (doctors) and 17 nurses | • General population |
| Nagykaldi et al[35] USA | • Uncontrolled observational | • Online Customised practice website including a bilingual influenza self-triage module, a downloadable influenza toolkit and electronic messaging capability. A bilingual seasonal influenza telephone hotline was available as an alternative. | • None | • Specific condition(s) Influenza |
| Nijland et al[20] Netherlands | • Uncontrolled observational Retrospective analysis of 15 months' data | • Online Web-based triage system (http://www.dokterdokter.nl) | • None | • General population |

Continued

**Table 1** Continued

| Reference | Study design | System type | Comparator | Population/sample |
|---|---|---|---|---|
| Poote et al[18] UK | • Uncontrolled observational | • Online Prototype self-assessment triage system | • Health professional performance on real-world data GPs triage rating was compared with rating from the self-assessment system | • General population Students attending a University Student Health Centre with new acute symptoms |
| Price et al[22] USA | • Uncontrolled observational | • Online A web-based decision support tool—SORT for kids designed to help parents and adult caregivers decide whether a child with possible influenza symptoms needs to visit the emergency department (ED) for immediate care. | • Health professional performance on real-world data The sensitivity of the algorithm was compared with a gold standard—evidence form child's medical records that they received one or more of five ED-specific interventions. | • Specific condition(s) Influenza in children |
| Semigran et al[6] N/A | • Experimental Described as an audit study | • Multiple 23 symptom checkers were evaluated. Symptom checkers available as apps (via the App Store and Google Play) were identified through searching for 'symptom checker' and 'medical diagnosis' and screened the first 240 results. Symptom checkers available online were identified through searching Google and Google Scholar for symptom checker and medical diagnosis and screened the first 300 results. | • Other Vignettes had a diagnosis and triage attached to them and these were compared against the symptom checker advice. | • General population Where a single class of illness was examined by the symptom checker, the symptom checker was excluded from the study. |
| Semigran et al[21] USA | • Experimental Comparison of physician and symptom checker diagnoses based on clinical vignettes | • Multiple 'Human Dx is a web-based and app-based platform' | • Health professional performance on test/ simulation Clincial vignettes—a comparison of 23 symptom checkers with physician diagnosis for 45 vignettes | • General population Of the 45 condition vignettes—there were 15 low, 15 medium and 15 high acuity vignettes—there were 26 common and 19 uncommon condition vignettes |
| Sole et al[19] USA | • Uncontrolled observational Descriptive comparative study | • Online A web-based triage system (24/7 WebMed) | • Health professional performance on real-world data Data were evaluated from students who had used the web-based triage and then requested an appointment via email (so triage data were available for comparison). | • General population |
| Yardley et al[14] UK | • Experimental Exploratory randomised trial | • Online Internet Doctor website | • Other Self-care information provided as a static web page with no symptom checker or triage advice | • Specific condition(s) Minor respiratory symptoms, for example, cough, sore throat, fever, runny nose |

GP, general practitioner; N/A, not applicable; NHS, National Health Service.

consultation or e-consultation). Study characteristics are summarised in table 2. In this type of system, the role of symptom checkers is to help patients decide whether their symptoms require a consultation with a doctor or other health professional or can be dealt with by self-care. If a consultation is required, details of the symptoms and a request for an appointment or call-back can be submitted electronically. This type of study is important because it considers the service within the broader context of the urgent and emergency care system. A limitation is that some studies focused mainly on the 'downstream' elements of the pathway, for example, consultation with

**Table 2** Studies of symptom checkers as part of an electronic consultation system

| Reference | Study design | System type | Comparator | Population/sample |
|---|---|---|---|---|
| Carter et al[24] UK | • Uncontrolled observational Mixed-method evaluation | • Online webGP (subsequently known as eConsult) | • Other Investigate patient experience by surveying patients who had used webGP and comparing their experience with controls (patients who had received a face-to-face consultation during the same time period) matched for age and gender | • General population General practices in NHS Northern, Eastern and Western Devon Clinical Commissioning Group's area |
| Cowie et al[25] UK | • Uncontrolled observational 6-month evaluation at 11 GP practices in Scotland | • Online eConsult, accessed via GP surgery websites. Service provides self-care assessment and advice, including symptom checkers; triage and signposting to alternative services; access to NHS24 (phone service) and e-consults allowing submission of details by email) | • None | • General population Patients registered with participating GP practices |
| Madan[26] UK | • Uncontrolled observational Report of 6-month pilot study | • Online webGP (subsequently known as eConsult) | • None | • General population |
| NHS England[7] UK | • Uncontrolled observational Analysis of data from four pilot studies together with data from other sources | • Multiple Pilots featured NHS pathways (web based; West Yorkshire); Sense.ly ('voice-activated avatar'; West Midlands); Espert 24 (Web-based; Suffolk) and babylon (app; North Central London) | • None Authors stated that it was not appropriate to compare pilot sites because of differences in starting date, 'footprints' covered, method of uptake and underlying population | • General population |
| Nijland et al[27] Netherlands | • Other Online survey | • Online Responses of interest relate to 'indirect e-consultation' (consulting a GP via secure email with intervention of a web-based triage system) | • None | • General population Patients with Internet access but no experience of e-consultation |

GP, general practitioner; NHS, National Health Service.

GPs, and provided limited data on the symptom checker element of the system.

A final group of five studies examined patient and/or public attitudes to online self-diagnosis in the context of urgent care.[28–32] See the full report for further details.[9]

## RESULTS BY OUTCOME
### Safety
None of the six included studies that reported on safety outcomes identified any problems or differences in outcomes between symptom checkers and health professionals. Most of the studies compared system performance with that of health professionals using real or simulated data. The only study with no comparison group was the 6-month pilot study of webGP,[26] which reported 'no major incidents'.

Limitations of the studies included not being based on real patient data[17], covering only a limited range of conditions[12 22] and sampling a young healthy population (students) not representative of the general population of users of the urgent care system.[18] Studies of e-consultation

systems did not generally collect data on those respondents who decided not to seek an appointment, limiting their ability to assess any impact on safety for this group. Overall, the evidence should be interpreted cautiously as indicating no evidence of a detrimental impact on safety rather than evidence of no detrimental effect.

### Clinical effectiveness
Only two studies reported on clinical effectiveness outcomes, making it difficult to draw any firm conclusions. In the study by Little et al, those who used the Internet Doctor website experienced longer illness duration and more days of illness rated moderately bad or worse than the usual care group.[13] The pilot study of the webGP system[26] reported that several patients received advice to seek treatment for serious symptoms that might otherwise have been ignored. However, no details or quantitative data were provided.

### Costs/cost effectiveness
Two included studies provided limited data on possible cost savings. Based on 6 months of pilot data,

Madan[26] estimated savings of £11 000 annually for an average general practice (6500 patients) compared with current practice. The report also suggested a saving to commissioners equivalent to £414 000 annually for a Clinical Commissioning Group (CCG, responsible for specifying and purchasing most health services in the NHS in England) covering 250 000 patients. These savings were specifically related to self-reported diversion of patients from GP appointments to self-care and from urgent care to e-consultation. Using similar methodology, the manufacturers of the babylon check app claimed average savings of over £10/triage compared with NHS111 by telephone, based on a higher proportion of patients being recommended to self-care.[23]

### Diagnostic accuracy

Eight studies reported at least some data on the diagnostic accuracy of symptom checkers. In spite of the diverse methods and comparisons in the included studies, almost all agreed that the diagnostic accuracy of symptom checkers was poor in absolute terms (eg, in evaluating 'vignettes' designed to test knowledge of specific conditions, where the correct diagnosis was already known by definition) or relative to that of health professionals. In the most comprehensive evaluation, Semigran et al evaluated 23 symptom checkers across 770 standardised patient evaluations.[6] Overall, the correct diagnosis was made in 34% of cases (95% CI 31% to 37%), although performance varied widely between symptom checkers, high and low acuity conditions and common and rare conditions. When the same authors compared the 23 symptom checkers with physicians using 43 vignettes, physicians were more likely to list the correct diagnosis first (out of three differential diagnoses) (72.1% vs 34% p<0.001) as well as among the top three diagnoses (84.3% vs 51.2% p<0.001).[21]

The only exception to the rule was an evaluation carried out at a student health centre.[19] Using data from 59 participants who used the 24/7 WebMed system and who were subsequently treated at the health centre, the study found good agreement between chief complaint, 24/7 WebMed classification and provider diagnosis (kappa values of 0.89–0.94). This study differed from the others in using data from students rather than a general population sample. In addition, the students' complaints were generally common and uncomplicated, a scenario in which symptom checkers performed relatively well in the study by Semigran et al.[21]

### Accuracy of disposition (triage and signposting to appropriate services)

Six included studies reported on this outcome, all except one of which[16] evaluated a 'general purpose' symptom checker. As with diagnostic accuracy, diverse methodologies and outcome measures were used.

The results overall presented a mixed picture but most studies indicated that symptom checkers were inferior and/or more cautious in their triage advice compared

with doctors or other health professionals. In their review of 23 symptom checkers, Semigran et al found that the systems provided appropriate triage advice in 57% (95% CI 52% to 61%) of cases.[6] Performance varied across the systems evaluated, correct triage ranging from 33% to 78%. The NHS England pilot evaluation of four systems[7] found that agreement with clinical experts varied from 30% to 95%, although the number of responses also varied, reducing the comparability of the results.

For abdominal pain, Berry et al evaluated three symptom checkers and found that 33% of diagnoses were at the same level of urgency as physician diagnoses (emergency, non-emergency or self-care); 39% were diagnosed as more serious and 30% less serious than the physician's judgement.[16] A similar level of agreement between algorithm and clinician (39%) was reported by Poote et al,[18] while the system evaluated by Nijland et al advised patients to visit a doctor in 85% of cases, even when the symptoms were appropriate for self-care.[20]

The only studies to report clearly equal or superior accuracy of disposition using an automated system were the evaluations of Babylon check by the company that developed the system. Middleton et al[17] reported that using patient vignettes, the app gave an accurate triage outcome in 88.2% of cases, compared with 75.5% for doctors and 73.5% for nurses (unaware of the 'correct' diagnosis for the vignettes). When vignettes were delivered by a medical professional rather than actors, the accuracy of Babylon check increased to over 90%. A later report looked at triage results obtained as part of the NHS England pilot evaluation, concluding that all of 74 referrals to urgent or emergency care were appropriate.[23]

### Impact on service use/diversion

Eight studies reported on this outcome, although one of them[12] merely stated that it was not possible to assess the effect of the intervention (a web-based influenza triage system) on patients' use of health services.

The pilot evaluation of the webGP system reported that 18% of users planned to book an appointment but chose not to do so.[26] In addition, 14% of users reported that they would have attended a walk-in centre or other urgent care service if they had not had access to the webGP system.

The NHS England pilot evaluation of four online/digital systems in different regions of England[7] compared the recommendations of the digital systems with those of the NHS111 telephone service over a similar time period (the first months of 2017). Compared with the telephone service, the online and digital services directed a slightly higher proportion of patients to self-care (18% vs 14%) and a lower proportion to other primary care services such as GPs, dental and pharmacy (40 vs 60%). The manufacturer's data on the babylon check app collected as part of the NHS England evaluation indicated that patients were more likely to be triaged to self-care by the app compared with NHS111 by telephone (40% vs 14%).[23] This figure includes people who received information leaflets on self-care as well as those who were actively triaged. If the

former group is excluded, the figures for the two services are similar (14% for NHS111% and 15.6% for babylon check[23]).

In their study of self-assessment for students attending a university health centre, Poote *et al* found that the prototype system they studied was able to identify a proportion of cases that doctors considered appropriate for self-care, suggesting a potential to reduce service use.[18] Similarly, Little *et al*'s RCT of a web-based symptom checker designed to support self-care for respiratory symptoms[13] reported that patients in the intervention group had fewer contacts with doctors than the usual care control group despite having a longer duration of illness and more days with relatively severe symptoms. This was balanced by an increase in contacts with the NHS Direct telephone service (which preceded NHS 111) and it should be noted that the system under evaluation recommended people needing treatment to contact NHS Direct rather than go directly to a doctor. Finally, a study of young adults (students) found that intention to seek treatment for a hypothetical illness was stronger when the diagnosis was made with the aid of WebMD or Google than with no electronic aid.[31]

### Patient compliance with triage advice

Only two of the included studies reported specifically on patients' compliance (or intention to comply) with advice received. The NHS England pilot evaluation in four regions asked participants in two of those regions (Suffolk and London) what they intended to do based on the advice received.[7] No quantitative data were provided but the report stated that in the Suffolk pilot, 'overall users would have followed the advice given'. However, those who were recommended to call 999 or attend an ED were more likely to seek advice from primary care or self-management. Similarly, in the London region, there was generally good agreement between advice and intended action but patients recommended to call 999 or go to an ED indicated that they would seek advice from a GP. In a study of a web-based triage system in the Netherlands, 192 patients were asked about their intention to comply immediately after receiving advice from the system.[20] Thirty-five patients responded to a follow-up survey on actual compliance, of whom 20 (57%) reported that they had followed the advice. Compliance was correlated with intention to comply, which in turn was correlated with the patient's attitude towards the advice received.

### Equity and inclusion

Fourteen studies investigated the outcome of equity and inclusion or compared users and non-users. One study[13] reported that patients who were classed as less deprived were more likely to agree to use 'Internet Doctor' than decline participation, although no relationship was found between deprivation and results in this study or between e-Consult use and deprivation in another study.[25] Association between e-consultation use and education levels was explored in a third study.

Patients with low to medium levels of education tended to be motivated toward indirect e-consultation (which involves contact with a health professional via email), mainly to reduce uncertainty.[27]

Evidence from included studies suggests that users of e-consultation were more likely to be young,[7 24–26] employed[20 24 26] and female[7 20 25 26] than non-users. One study also found a significantly larger use by white patients (78%) than other ethnicities.[25]

### Risk-of-bias assessment

We assessed risk of bias in the two included RCTs[13 14] using the Cochrane risk of bias tool. Thirteen studies[12 19 20 24 25 27–30 32–35] were assessed with the tool for cross-sectional and cohort studies and four (six publications[6 18 21 22 36 37]) with the modified QUADAS tool. Seven grey literature reports and conference abstracts were not formally assessed for risk of bias.[7 15–17 26 31 38] Identified limitations were extracted for all included studies.

Risk-of-bias results are presented in online supplementary appendix 2. With the possible exception of the two randomised trials, the included studies generally had at least a moderate risk of bias. However, the diverse designs and objectives of the studies made risk of bias difficult to assess in some cases with the available tools. Grey literature reports containing relevant data were included in the review but not formally assessed for risk of bias. Reports prepared by individuals with a commercial interest in a specific system and published without independent peer review[17 26] should be treated with particular caution because of possible conflicts of interest.

### Overall strength of evidence assessment/evidence map

The overall strength of evidence for key outcomes is summarised in table 3. We found relatively strong evidence that the diagnostic accuracy of digital and online symptom checkers tends to be lower than that of health professionals; and that patients who have used these systems generally show high levels of satisfaction (mainly in non-comparative studies). Areas where evidence is lacking or inconsistent include clinical and cost effectiveness, accuracy of disposition to appropriate services and patient compliance with advice received. For safety, we found no evidence of an increased risk with digital/online systems but the available evidence was weak.

### DISCUSSION
### Main findings

The literature search identified 29 publications describing 27 studies that met the inclusion criteria. The overall strength of the evidence base varied between outcomes (table 3), but in absolute terms the evidence is weak, being based largely on observational studies. A substantial component of grey literature of uncertain quality complicates the interpretation of the evidence. Interpretation of the evidence should also take into account risks of bias in individual studies. In addition, one included study

**Table 3** Overall strength of evidence by outcome

| Outcome | Relevant studies | Evidence statement | Strength of evidence | Comments |
|---|---|---|---|---|
| Safety | =Kellermann et al[12]<br>=Little et al[13]<br>=Middleton et al[17]<br>=Poote et al[18]<br>=Price et al[22]<br>Madan[26] | No evidence of a difference in risk between health professionals and symptom checkers | Weaker | Rating changed from stronger based on study numbers and design to weaker because of low numbers of adverse events reported |
| Clinical effectiveness | –Little et al[13]<br>?Madan[26] | Insufficient evidence to draw any firm conclusions | Very limited | |
| Costs/<br>cost effectiveness | +Babylon Health[23]<br>±Cowie et al[25]<br>+Madan[26] | Insufficient evidence to draw any firm conclusions | Inconsistent | |
| Diagnostic accuracy | ?Berry et al[15]<br>–Berry et al[38]<br>– Berry et al[16]<br>– Price et al[22]<br>?Semigran et al[6]<br>–Semigran et al[21]<br>=Sole et al[19] | Symptom checkers appear inferior to health professionals in terms of diagnostic accuracy | Stronger | Mainly for specific conditions or preprepared vignettes |
| Disposition accuracy | =Babylon Health[23]<br>–Berry et al[16]<br>=Middleton et al[17]<br>?Nijland et al[20]<br>– Poote et al[18]<br>±Semigran et al[6]<br>±NHS England[7] | Inconsistent findings on accuracy of disposition | Inconsistent | Performance variable between different systems |
| Service use/diversion | ?Kellermann et al[12]<br>±Little et al[13]<br>±Poote et al[18]<br>?Carter et al[24]<br>?Cowie et al[25]<br>+Madan[26]<br>±NHS England[7]<br>+Babylon Health[23]<br>–Luger et al[33] | Inconsistent findings on effects on service use | Inconsistent | |
| Compliance | ?Nijland et al[20]<br>?NHS England[7] | No comparative data on compliance | Very limited | |
| Patient/carer satisfaction | ?Nagykaldi et al[35]<br>?Nijland et al[20]<br>?Price et al[22]<br>+Yardley et al[14]<br>?Carter et al[24]<br>?Cowie et al[25]<br>?Madan[26]<br>?NHS England[7]<br>?Lanseng and Andreassen[30] | Most studies report high rates of patient satisfaction with symptom checkers and e-consultation systems generally | Weaker | Few studies with comparator data |

Controlled studies in bold; =means no significant difference in outcomes; +means better outcome with symptom checker; ±varying results within study; ? results difficult to interpret in comparative terms.

evaluated 23 symptom checkers and only the overall findings are summarised in this review.[6]

We found little evidence to indicate whether or not digital and online symptom checkers are detrimental to patient safety. The studies that reported on the outcome were mostly short term and involved relatively small samples and hence reported few or no adverse events. Some were limited to people with specific types of symptoms and others recruited from specific population groups not representative of typical users of urgent care services. This body of evidence should therefore be interpreted cautiously and not extrapolated to the possible impact of a nationally available digital urgent care service being used by millions of people annually.

The evidence on patient satisfaction with digital and online systems also had some limitations but these findings appear more likely to be generalisable. Study participants generally expressed high levels of satisfaction, although in uncontrolled studies. For example, in the NHS England pilot evaluation, 70%–80% of users were satisfied with their experience at each of the pilot sites.[7] This evidence, together with the increasing reliance on digital technology in all areas of life, suggests that any national digital urgent care service may be popular and

well used, although different sections of the population may differ in their degree of engagement (see the discussion of equity and inclusion below).

Digital and online systems have yet to achieve a high level of accuracy in the diagnosis of specific conditions. This finding applies both to 'general purpose' symptom checkers and to those limited to particular conditions. Although the evidence was classified as relatively strong, several caveats should be applied. Some of the included studies did not recruit representative populations and others were based on standardised vignettes rather than real-world data. In addition, studies that compared symptom checkers with health professionals tended to use the doctors' clinical diagnosis as the reference standard, which would bias the comparison in favour of the health professionals. Poor diagnostic accuracy could also have implications for patient safety, although the limited evidence on safety outcomes (small samples and small numbers of events) makes it difficult to draw any firm conclusions. If symptom checkers are generally risk averse, this could potentially mitigate any effects on safety.

Accuracy of signposting of patients to the most appropriate level of service is closely related to diagnostic accuracy, but results for this outcome were inconsistent between studies. In general, algorithm-based triage tended to be more risk averse than that of health professionals, with 85% of respondents being advised to visit their doctor in one study.[20] While there is considerable uncertainty about the magnitude of the effect, a national digital urgent care service could result in considerable numbers of patients receiving inappropriate advice to visit the ED or request an urgent GP appointment. Middleton *et al*[17] claimed that the babylon check app had a high degree of triage accuracy for vignettes compared with health professionals, but this non-peer-reviewed report requires further validation.

We also found inconsistent evidence on effects on service use. There was some indication that symptom checkers can influence the pattern of service use but the magnitude and direction of the effect varied between studies. Patients' reactions to online triage advice and whether they follow the advice or seek further help or information would have implications for service use but we found limited evidence for this outcome. Preliminary findings from the NHS England evaluation suggest that patients may be more likely to seek further advice for more urgent conditions[7] but further confirmation is required.

Over half of the included studies considered equity and inclusion issues either directly or by comparing users and non-users of digital triage systems. Not surprisingly, studies revealed a clear consensus that younger and more highly educated people are more likely to use these services while older and less educated patients are more likely to prefer telephone or face-to-face contact. This could have implications for health equity if urgent care pathways prioritise (or appear to prioritise) requests originating from digital sources. Problems have arisen in primary care because patients using e-consultation systems to request an appointment following online triage may be seen more quickly than those contacting the practice by telephone.

## Strengths and limitations

This systematic review was undertaken on a short timescale using a relatively large team of experienced researchers, including both methodological and topic experts. We performed a rigorous search of the literature including reference checking and citation searching. Rather than a conventional highly sensitive search (which would have resulted in inefficiencies in the screening process), we combined an initial focused search with subsequent rounds of follow-up searching, including searches for named symptom checker systems. We assessed risk of bias in individual studies using a variety of appropriate checklists as well as summarising the overall strength of evidence for key outcomes (table 3).

The heterogeneous and descriptive nature of the included studies meant that meta-analysis was not feasible for any of the outcomes of interest. Our narrative synthesis approach used a mixture of description and tabulation to summarise the evidence for each of the prespecified outcomes of interest. This was a review of published (including non-peer-reviewed) literature and the coverage of systems is not exhaustive; for example, we did not extract data from websites. We also did not carry out any original analyses of raw data even where such data were available. The timing of the review meant that final results of NHS England's pilot evaluation were not available to us. We were able to make use of a draft report that was published online[7] but we acknowledge that the findings of the final evaluation report, when available, will supersede those of the 2017 draft.

The review inclusion criteria were relatively broad and findings from symptom checker systems for specific conditions may not be applicable to more general systems and vice versa. We have also included studies of symptom checkers as part of electronic consultation systems in general practice, which again represents a slightly different setting from a general digital 111 service, and this should be kept in mind when interpreting the results.

A systematic review in such a topical area of research will require regular updating to keep track of new studies. For example, Verzantvoort *et al*[39] published a study of self-triage using a smartphone app for out-of-hours primary care in the Netherlands shortly after our literature searches were completed. The app was rated highly for clarity and patient satisfaction. Sensitivity and specificity (using nurse telephone triage as reference standard) were 84% and 74% respectively, although diagnostic accuracy was only evaluated in a sample of participants (126/4456). Inclusion of this study would not have affected the main conclusions of our review.

## Implications for service delivery and research

The implications of this systematic review for service delivery should be considered in the context that a

decision has already been taken to introduce a digital 111 service and the service became available across England by December 2018. Achieving a high level of diagnostic accuracy will be key to the success of a digital 111 service. Failure to provide an accurate diagnosis may result in outcomes including patient dissatisfaction and unwillingness to use the service again; increased use of other urgent and emergency care services and possible risks to patient safety (although the cautious approach characteristic of most existing systems may help to mitigate this).

The studies included in the review suggest a high level of uncertainty about the impact of digital 111 on the urgent care system and the wider healthcare system. Some of these uncertainties can be addressed by research and data collection but the health service may need to respond to short-term increases (or decreases) in demand and/or shifts from one part of the system to another. This may increase pressure on the system, at least in the short term. In the longer term, if usage of the 111 telephone service decreases as planned, there may be opportunities to reconfigure the workforce to support the integrated urgent care agenda.

Based on the areas of limited evidence identified by the review, priorities for research (in addition to ongoing collection of data to monitor usage and safety of the digital 111 service) include studies to compare the performance of different systems directly; rigorous economic evaluations based on real-world data; research to investigate the pathways followed by patients using the service; evaluation of systems designed for childhood illnesses and investigation of the possible role of behavioural change theory in the development and implementation of symptom checkers. Qualitative research to investigate perceptions of symptom checkers and barriers to their use by people who are less familiar with digital technology would also be of value.

**Contributors** DC contributed to the planning (project coordination and protocol development), conduct (study selection, data extraction and quality assessment) and reporting (report writing) of the study. AJC contributed to the planning (protocol development), conduct (information retrieval, study selection, data extraction and quality assessment) and reporting (report writing) of the study. MJ, LP and SKB contributed to the planning (protocol development), conduct (study selection, data extraction and quality assessment) and reporting (report writing) of the study. AB contributed to the planning (protocol development), conduct (information retrieval and study selection) and reporting (report writing) of the study. JT contributed to the planning, conduct and reporting of the study by providing expert topic advice at all stages. All the authors contributed to the study conception and design (protocol development), acquisition of data (study selection and data extraction) and analysis or interpretation of data (writing sections and/or commenting on drafts of the report). DC is the guarantor for this work. The corresponding author attests that all listed authors meet authorship criteria and that no others meeting the criteria have been omitted.

**Funding** This report presents independent research funded by the National Institute for Health Research (NIHR) Health Services & Delivery Research Programme (project number HSDR16/47/17). The funding programme approved the review protocol but had no role in the collection, analysis and interpretation of the data, the writing of this paper or the decision to submit the paper for publication.

**Disclaimer** The views and opinions expressed are those of the authors and do not necessarily reflect those of the NHS, the NIHR, NETSCC, the HS&DR programme or the Department of Health.

**Competing interests** None declared.

**Patient consent for publication** Not required.

**Provenance and peer review** Not commissioned; externally peer reviewed.

**Data sharing statement** No new data have been created in the preparation of this report and therefore there is nothing available for access and further sharing. All queries should be submitted to the corresponding author.

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
