## [Reviewer comments · BMJ Open]

ARTICLE DETAILS

TITLE (PROVISIONAL)	Digital and online symptom checkers and health assessment/triage services for urgent health problems: systematic review
AUTHORS	Chambers, Duncan; Cantrell, Anna; Johnson, Maxine; Preston, Louise; Baxter, Susan; Booth, Andrew; Turner, Janette

VERSION 1 - REVIEW

REVIEWER	Caroline Jay Senior Lecturer in Computer Science University of Manchester, UK
REVIEW RETURNED	26-Nov-2018

GENERAL COMMENTS	This is an interesting and timely review. The synthesis of the findings appears justified and useful. The method would benefit from further description in a few areas: It is common for queries within a systematic review to yield large numbers of results, which then require sifting. What does 'disproportionately high' (p.6) mean in concrete terms? Does the search strategy shown in Appendix 1 show precisely how articles were retrieved from MEDLINE? It would be good to detail the other focused search strategies for completeness, as the description of how articles were retrieved is vague at present. There are a number of points where a 10% sample was checked for accuracy and consistency. How was the sample determined? What were the results of these checks? How much disagreement was there? p.9 At present, it isn't completely clear how the PPI fed into the paper. For example, what were the potential concerns, and how were they addressed in the final draft of the review (if they were)? What were the highlighted issues and how were they used to structure the findings? p.8 Add references for the QUADAS and National Heart and Lung Institute tools. On p. 8 and p.10 reference is made to a 'full report (Chambers et al in preparation) but this does not appear in the references and is not accessible.
--

REVIEWER	Alike van der Velden Julius Center for Health Sciences and Primary Care, University Medical Center Utrecht, the Netherlands
REVIEW RETURNED	24-Feb-2019

GENERAL COMMENTS	The rapidly increasing number of online tools to aid patients in investigating their own health and introduction of these tools by the formal health care system, urgently asks for a proper evaluation of safety and effectiveness of these systems, as was done by the authors in this review. The aim, outcomes they assessed are clear, and the results highlight the need for more, not only observational, but controlled research in this area. I've some comments, which the authors might want to include, or explain: - Introduction:  1) focusses a bit too much on the UK NHS111 service (STPs unclear) 2) why is there a risk of duplicating healthcare contacts? 3) I would expect these tools to reduce from urgent to non-urgent care, or phone calls in general, not reducing the emergency calls/care. Were these evaluated? 4) Would be helpful for reader to add that ref 5 used patient vignettes. - Methods:  1) Relevant to mention the timing of their literature search, up to beginning 2018? 2) In the selected studies I miss the following study, evaluating a self-triage app for out-of-hours (urgent) care in the Netherlands: PLoS One. 2018; 13(6): e0199284. Self-triage for acute primary care via a smartphone application: Practical, safe and efficient? N. Verzantvoort, T. Teunis, T. Verheij, A. van der Velden. It might be that the literature search was done in the beginning of 2018. Nevertheless, it might be relevant to mention this study in the Discussion, or include in the analysis. 3) 10% of checks for data extraction by a second reviewer, is actually 2 or 3 studies, not many for a check. 4) Can the authors provide a definition for 'grey literature'? - Results:  1) In the methods they describe the PPI group, of which I miss any results, or discussion in the remainder of the manuscript. Would be interesting to read about their messages 2) Would be interesting to see the countries where the studies were performed in the Tables. 3) I've got problems with study no 5 mentioned as one in the Table. That study was a test of 23 individual symptom checkers, with some very bad and a few comparatively well performing. Now they are presented as 'one result'. Why not evaluating the individual checkers? 4) It's hard to believe that 'none of the 6 included studies that reported on safety outcomes identified any problems or differences in outcomes between checkers and HC professionals', also in the light of the review of Semigran. Or there is no issue with safety when they are all risk-averse. 5) For non-UK readers, what is a CCG? 6) The results on diagnostic accuracy (poor in absolute terms) conflicts, or partly conflicts with safety results. 7) Can the authors explain which standard was used in the study number 14? If it wasn't a doctor, who/what was it?
--

	8) When it is 'unclear whether it was dealt with real or hypothetical data' and 'overall users would have followed advice given', these results are meaningless to me (page 20 and 21). - Discussion: 1) Overall message is clear 2) Would like to see some personal thoughts/results from PPI and introduction of the online digital 111 service
--	---

REVIEWER	Sean Ewings University of Southampton United Kingdom
REVIEW RETURNED	06-Mar-2019

GENERAL COMMENTS	I think this is a very good study that is well written, and which could be accepted without revision. The abstract was clear, the rationale, methods and results were presented clearly, and the conclusions were in line with the results. I had only minor comments and offer these as thoughts for the team to consider. 1. It was interesting to hear the benefits of using PPI, though it would be interesting to also understand what concerns were raised and how this was dealt with - at the moment this is slightly vague. 2. Paragraph 3 of Discussion: I thought the point that digital services "may be popular and well-used" was slightly hasty given that you later say that age would affect engagement. I was not clear if the high levels of satisfaction would have been inflated by this only being measured in people who use the digital services - hence this represents a characteristic of the digital platform, which, while useful, does not represent satisfaction with services as a whole (i.e., it does not account for the potential of non-digital users who may experience delays, as you suggest, as a result of digital users being prioritised after online triage - I appreciate this would be hard for any study to measure though!). 3. The Babylon check system seemed to perform well with regards to accuracy of disposition. Whilst you carefully acknowledge the role of the developers in the assessment of this system, it would be interesting to know if you felt the vignettes used (or perhaps another part of the assessment process) was particularly different compared to other studies that did not report such high rates of accuracy.
--

VERSION 1 – AUTHOR RESPONSE

Reviewer(s)' Comments to Author:

Reviewer: 1

Reviewer Name: Caroline Jay

Institution and Country: Senior Lecturer in Computer Science, University of Manchester, UK

Please state any competing interests or state 'None declared': None declared.

Please leave your comments for the authors below

This is an interesting and timely review. The synthesis of the findings appears justified and useful.
Thank you

The method would benefit from further description in a few areas:

It is common for queries within a systematic review to yield large numbers of results, which then require sifting. What does 'disproportionately high' (p.6) mean in concrete terms? This was not quantified but examination of sample search results showed that the strategy was not retrieving the types of papers we were interested in. As the review had a short time frame, we wanted to streamline the sifting process as much as possible.

Does the search strategy shown in Appendix 1 show precisely how articles were retrieved from MEDLINE? It would be good to detail the other focused search strategies for completeness, as the description of how articles were retrieved is vague at present. We have revised Appendix 1 to include search strategies for all databases

There are a number of points where a 10% sample was checked for accuracy and consistency. How was the sample determined? What were the results of these checks? How much disagreement was there? This was a random sample and we have added 'random' to the methods where appropriate. We have added the inter-rater agreement (kappa) for study selection (pp 9-10). Disagreements on DE/QA were resolved between reviewers and not formally quantified.

p.9 At present, it isn't completely clear how the PPI fed into the paper. For example, what were the potential concerns, and how were they addressed in the final draft of the review (if they were)? What were the highlighted issues and how were they used to structure the findings? We have added a few sentences to clarify this point (p9).

p.8 Add references for the QUADAS and National Heart and Lung Institute tools. Added

On p. 8 and p.10 reference is made to a 'full report (Chambers et al in preparation) but this does not appear in the references and is not accessible. Now accepted for publication; we have added details to the text reference

Reviewer: 2

Reviewer Name: Alike van der Velden

Institution and Country: Julius Center for Health Sciences and Primary Care, University Medical Center Utrecht, the Netherlands

Please state any competing interests or state 'None declared': None declared

Please leave your comments for the authors below

The rapidly increasing number of online tools to aid patients in investigating their own health and introduction of these tools by the formal health care system, urgently asks for a proper evaluation of safety and effectiveness of these systems, as was done by the authors in this review. The aim, outcomes they assessed are clear, and the results highlight the need for more, not only observational, but controlled research in this area.

I've some comments, which the authors might want to include, or explain:

- Introduction:

- 1) focusses a bit too much on the UK NHS111 service (STPs unclear) We have added a little more explanation and a reference for STPs (p5)
- 2) why is there a risk of duplicating healthcare contacts? Added brief explanation (p5)
- 3) I would expect these tools to reduce from urgent to non-urgent care, or phone calls in general, not reducing the emergency calls/care. Were these evaluated? The 111 telephone service was introduced to reduce pressure on the 999 emergency service and this is what was evaluated in the study cited (from 2013)
- 4) Would be helpful for reader to add that ref 5 used patient vignettes. Done

- Methods:

- 1) Relevant to mention the timing of their literature search, up to beginning 2018? We have added that the main literature search was conducted in April 2018
- 2) In the selected studies I miss the following study, evaluating a self-triage app for out-of-hours (urgent) care in the Netherlands: PLoS One. 2018; 13(6): e0199284. Self-triage for acute primary care via a smartphone application: Practical, safe and efficient? N. Verzantvoort, T. Teunis, T. Verheij, A. van der Velden. It might be that the literature search was done in the beginning of 2018. Nevertheless, it might be relevant to mention this study in the Discussion, or include in the analysis. This study was published in June 2018 after our literature search was completed. We have added a reference to the study in the discussion.
- 3) 10% of checks for data extraction by a second reviewer, is actually 2 or 3 studies, not many for a check. Acknowledged, but supplemented by informal checking while writing the technical report and paper, and minor data extraction errors unlikely to affect review findings
- 4) Can the authors provide a definition for 'grey literature'? Added (p8)

- Results:

- 1) In the methods they describe the PPI group, of which I miss any results, or discussion in the remainder of the manuscript. Would be interesting to read about their messages. We have added some text to the PPI section (p9)
- 2) Would be interesting to see the countries where the studies were performed in the Tables. Not extracted for the review and not of a lot of value as many of the studies were independent of country/health system

3) I've got problems with study no 5 mentioned as one in the Table. That study was a test of 23 individual symptom checkers, with some very bad and a few comparatively well performing. Now they are presented as 'one result'. Why not evaluating the individual checkers? For the review, we summarised the findings of the study as a whole. Details of the individual symptom checkers can be found in the original publication

4) It's hard to believe that 'none of the 6 included studies that reported on safety outcomes identified any problems or differences in outcomes between checkers and HC professionals', also in the light of the review of Semigran. Or there is no issue with safety when they are all risk-averse. It is probably a mixture of symptom checkers being risk averse and studies having small samples (hence small numbers of events) or not using real patient data (e.g. vignette studies)

5) For non-UK readers, what is a CCG? Explanation added (p18)

6) The results on diagnostic accuracy (poor in absolute terms) conflicts, or partly conflicts with safety results. The apparent conflict reflects the limited evidence on safety as mentioned above and highlighted in the discussion (p26)

7) Can the authors explain which standard was used in the study number 14? If it wasn't a doctor, who/what was it? This was a vignette study so the reference standard was the pre-specified correct diagnosis for the vignette. The symptom checker apparently performed well compared with clinicians who were unaware of the correct diagnosis. We have added a few words to clarify this (p20)

8) When it is 'unclear whether it was dealt with real or hypothetical data' and 'overall users would have followed advice given', these results are meaningless to me (page 20 and 21). This reflects the limitations of an unpublished draft report. We agree that the sentence about real and hypothetical cases is confusing and have removed it (p21)

- Discussion:

1) Overall message is clear. Thank you

2) Would like to see some personal thoughts/results from PPI and introduction of the online digital 111 service. We have added this to the PPI section as mentioned above

Reviewer: 3

Reviewer Name: Sean Ewings

Institution and Country: University of Southampton, United Kingdom

Please state any competing interests or state 'None declared': None declared.

Please leave your comments for the authors below

I think this is a very good study that is well written, and which could be accepted without revision. The abstract was clear, the rationale, methods and results were presented clearly, and the conclusions were in line with the results. I had only minor comments and offer these as thoughts for the team to consider. Thank you

1. It was interesting to hear the benefits of using PPI, though it would be interesting to also understand what concerns were raised and how this was dealt with - at the moment this is slightly vague. We have added this to the PPI section as mentioned above

2. Paragraph 3 of Discussion: I thought the point that digital services "may be popular and well-used" was slightly hasty given that you later say that age would affect engagement. I was not clear if the high levels of satisfaction would have been inflated by this only being measured in people who use the digital services - hence this represents a characteristic of the digital platform, which, while useful, does not represent satisfaction with services as a whole (i.e., it does not account for the potential of non-digital users who may experience delays, as you suggest, as a result of digital users being prioritised after online triage - I appreciate this would be hard for any study to measure though!). We have amended this section (p26) and also added a suggestion for more qualitative research into barriers to uptake.

3. The Babylon check system seemed to perform well with regards to accuracy of disposition. Whilst you carefully acknowledge the role of the developers in the assessment of this system, it would be interesting to know if you felt the vignettes used (or perhaps another part of the assessment process) was particularly different compared to other studies that did not report such high rates of accuracy. Unfortunately, the study was not published in a peer-reviewed journal so the data are not available to comment on this point.

VERSION 2 – REVIEW

REVIEWER	Caroline Jay University of Manchester, UK
REVIEW RETURNED	20-Apr-2019

GENERAL COMMENTS	The revised manuscript broadly addresses the issues raised in the previous review. A final question regarding the strategies in Appendix 1: are these precise descriptions, or examples? It would be useful to explain this in the manuscript from the perspective of reproducibility.
--

REVIEWER	Alike van der Velden Julius Center for Health Sciences and Primary Care University Medical Center Utrecht The Netherlands
REVIEW RETURNED	14-Apr-2019

GENERAL COMMENTS	Thank you for implementing the suggested changes. I feel that there are three issues that were responded to in the letter, which would need a few words in the Discussion as other readers might experience the same demur I had. These are: Results item 2: Some information about the countries where studies were performed. You mention in your letter that studies were performed independent of HC system. I think this is not true,
--

	every participant is part of a HC system and acts within the context of their national system and HC seeking behaviour. Results item 3: I would at least suggest to mention this in the Discussion that results of 23 checkers were combined into one overall result. This procedure might have shifted the weight in the reporting the outcomes. Results item 4/6: I would suggest mentioning this observed discrepancy explicitly/in more detail in the Discussion section.
--	--

REVIEWER	Sean Ewings University of Southampton, UK
REVIEW RETURNED	24-Apr-2019

GENERAL COMMENTS	You have addressed all of my comments and I would recommend this for publication.
---

VERSION 2 – AUTHOR RESPONSE

Reviewer(s)' Comments to Author:

Reviewer: 2

Reviewer Name: Alike van der Velden

Institution and Country: Julius Center for Health Sciences and Primary Care, University Medical Center Utrecht, The Netherlands

Please state any competing interests or state 'None declared': None declared

Please leave your comments for the authors below

Thank you for implementing the suggested changes.

I feel that there are three issues that were responded to in the letter, which would need a few words in the Discussion as other readers might experience the same demur I had.

These are:

Results item 2: Some information about the countries where studies were performed. You mention in your letter that studies were performed independent of HC system. I think this is not true, every participant is part of a HC system and acts within the context of their national system and HC seeking behaviour. We have added country to Tables 1 and 2 and the relevant section of the text

Results item 3: I would at least suggest to mention this in the Discussion that results of 23 checkers were combined into one overall result. This procedure might have shifted the weight in the reporting the outcomes. Added to the first paragraph of the discussion

Results item 4/6: I would suggest mentioning this observed discrepancy explicitly/in more detail in the Discussion section. Added some text to the Discussion (p27)

Reviewer: 1

Reviewer Name: Caroline Jay

Institution and Country: University of Manchester, UK

Please state any competing interests or state 'None declared': None declared.

Please leave your comments for the authors below

The revised manuscript broadly addresses the issues raised in the previous review. A final question regarding the strategies in Appendix 1: are these precise descriptions, or examples? It would be useful to explain this in the manuscript from the perspective of reproducibility. These are the exact search strategies implemented by our information specialist/reviewer (AC). We have modified the wording to make this clearer (p7)

Reviewer: 3

Reviewer Name: Sean Ewings

Institution and Country: University of Southampton, UK

Please state any competing interests or state 'None declared': None declared

Please leave your comments for the authors below

You have addressed all of my comments and I would recommend this for publication. Thank you!

VERSION 3 - REVIEW

REVIEWER	Caroline Jay University of Manchester, UK
REVIEW RETURNED	27-Jun-2019

GENERAL COMMENTS	The paper reads well and I am happy for it to be published.
---